# Retinoblastoma from human stem cell-derived retinal organoids

Jackie L. Norrie [1], Anjana Nityanandam[1], Karen Lai[1], Xiang Chen [2], Matthew Wilson[3], Elizabeth Stewart [1,4], Lyra Griffiths[1], Hongjian Jin[5], Gang Wu [5], Brent Orr [6,7], Quynh Tran [6,7], Sariah Allen[6,7], Colleen Reilly[2], Xin Zhou[2], Jiakun Zhang[1], Kyle Newman[1], Dianna Johnson[3], Rachel Brennan [4✉] & Michael A. Dyer [1,3,6,7✉]

Retinoblastoma is a childhood cancer of the developing retina that initiates with biallelic inactivation of the *RB1* gene. Children with germline mutations in *RB1* have a high likelihood of developing retinoblastoma and other malignancies later in life. Genetically engineered mouse models of retinoblastoma share some similarities with human retinoblastoma but there are differences in their cellular differentiation. To develop a laboratory model of human retinoblastoma formation, we make induced pluripotent stem cells (iPSCs) from 15 participants with germline *RB1* mutations. Each of the stem cell lines is validated, characterized and then differentiated into retina using a 3-dimensional organoid culture system. After 45 days in culture, the retinal organoids are dissociated and injected into the vitreous of eyes of immunocompromised mice to support retinoblastoma tumor growth. Retinoblastomas formed from retinal organoids made from patient-derived iPSCs have molecular, cellular and genomic features indistinguishable from human retinoblastomas. This model of human cancer based on patient-derived iPSCs with germline cancer predisposing mutations provides valuable insights into the cellular origins of this debilitating childhood disease as well as the mechanism of tumorigenesis following *RB1* gene inactivation.

[1] Department of Developmental Neurobiology, St. Jude Children's Research Hospital, Memphis, TN, USA. [2] Department of Computational Biology, St. Jude Children's Research Hospital, Memphis, TN, USA. [3] Department of Ophthalmology, University of Tennessee Health Science Center, Memphis, TN, USA. [4] Department of Oncology, St. Jude Children's Research Hospital, Memphis, TN, USA. [5] Center for Applied Bioinformatics, St. Jude Children's Research Hospital, Memphis, TN, USA. [6] Department of Pathology, St. Jude Children's Research Hospital, Memphis, TN, USA. [7] Howard Hughes Medical Institute, Chevy Chase, MD, USA. ✉email: rachel.brennan@stjude.org; michael.dyer@stjude.org

Retinoblastoma is a rare pediatric cancer of the developing retina that initiates in utero and is diagnosed within the first few years of life[1]. The vast majority of retinoblastomas initiate with biallelic inactivation of *RB1* and a small subset (1–2%) initiate with *MYCN* amplification in the absence of *RB1* inactivation[2,3]. Approximately half of all retinoblastoma cases involve a germline mutation in *RB1* and 25% of germline retinoblastoma patients inherited the mutant allele from a parent[4]. Patients with germline *RB1* mutations have an earlier age of onset because they only require inactivation of the remaining *RB1* allele for tumor initiation[1,5]. Genomic studies have indicated that *RB1* inactivation is sufficient for tumorigenesis[6]. Therefore, retinoblastoma is an important model of tumor initiation as a result of biallelic inactivation of a single tumor suppressor gene.

Early attempts to model retinoblastoma in mice by mutating the *Rb1* gene failed to produce retinal tumors in $Rb^{+/-}$ mice[7–9]. Subsequent conditional inactivation of both copies of *Rb1* in the developing murine retina also failed to produce retinoblastoma[10–13]. Further molecular and genetic studies demonstrated species-specific intrinsic genetic redundancies and compensation among Rb family members prevent retinoblastoma in mice[14]. Conditional inactivation of *Rb1* and *p107* or *Rb1* and *p130* can lead to retinoblastoma[10–14] in mice and more recently, a murine model of the *MYCN* amplified form of retinoblastoma was developed[15].

While these genetically engineered mouse models (GEMMs) have provided important insights into retinoblastoma biology, there are some differences in the molecular and cellular features of retinoblastoma across species[11,14,16,17]. For example, in a recent epigenetic study of the developing retina and retinoblastoma from mice and humans, the human tumor epigenome mapped to a later developmental stage than that of the mouse tumors[18]. Those differences may be related to the initiation of tumorigenesis during retinal development, the cellular origin of the tumors, or both[19]. Another important difference between human and murine retinoblastoma is their drug sensitivity. In side-by-side preclinical studies of murine retinoblastoma and orthotopic patient-derived xenografts (O-PDXs) of retinoblastoma, there were differences in the response to the conventional systemic chemotherapy regimen currently used to treat retinoblastoma patients[20]. Virtually, all of the GEMMs had a complete and durable response while few of the O-PDXs tumor-bearing mice had any response[20]. Therefore, a laboratory model of human retinoblastoma could contribute to our understanding of the retinoblastoma cell of origin and provide patient-specific tumors for preclinical testing.

To produce a laboratory model of human retinoblastoma, we created iPSCs from 15 participants with germline *RB1* mutations or deletion. Multiple clones from each iPSC line were validated for retention of the germline mutation, subjected to molecular profiling, including whole-genome sequencing, and characterized for embryoid body formation, neural rosette formation, and retinal differentiation using an improved human retinal organoid procedure. Representative clones from each participant were then differentiated into retinal organoids, dissociated, and injected into the eyes of immunocompromised mice to monitor tumor formation. Parallel experiments were performed with targeted *RB1* gene inactivation using CRISPR-Cas9. Individual tumors derived from the organoids were passaged by intraocular injection and cryopreserved, as done previously[21]. Complete molecular, cellular, and genetic profiling were completed and the iPSC lines and tumors have been made available to the biomedical research community free of charge with no obligation to collaborate through the Childhood Solid Tumor Network[21].

## Results

**Isolation and characterization of patient-derived iPSC lines.** In total, 11 patients and 4 family members were enrolled on the RETCELL (NCT02193724) protocol at St. Jude Children's Research Hospital (Supplementary Data 1 and Fig. 1A, B). Four samples for reprogramming were obtained by skin biopsy at the time of anesthesia for examination of the eyes as part of routine care, and eleven samples were obtained with the peripheral blood draw. Germline DNA samples were also obtained from 14 of the 15 participants. Participants were selected based on their clinical presentation to represent a broad spectrum of penetrance in heritable retinoblastoma disease. Four participants had no family history and were diagnosed at less than 12 months of age (bilateral, $n = 3$; asynchronous bilateral, $n = 1$) and two participants had 13q deletions (manifested as a unilateral and bilateral disease). Four family cohorts ($n = 9$) consented to the protocol, including two families where the parent was previously unknown to carry the *RB1* mutation until the children were diagnosed with bilateral or trilateral retinoblastoma (Supplementary Data 1 and Fig. 1A, B). Offspring of the family cohorts presented with bilateral retinoblastoma ($n = 4$) and trilateral disease ($n = 1$) at diagnosis, and the parents had no evidence of tumors by ophthalmologic screening.

Fibroblasts or blood were reprogrammed with the CytoTune Sendai reprogramming kit (Supplemental Information). In total, 71 iPSC clones were isolated for 15 participants and the germline DNA mutations in *RB1* were verified by Sanger sequencing for most lines, and fluorescent in situ hybridization for SJRB-iPSC-13 which had a *RB1* locus deletion (Supplementary Data 2, Fig. 1A–E and Supplemental Information). Only fully reprogrammed clones with trilineage differentiation potential and a normal karyotype were used for subsequent studies, hereafter referred to as SJRB-iPSC-1-15 (Supplementary Fig. 1A–H and Supplementary Data 2). Whole-genome sequencing was performed on the iPSC clones, and the patient germline DNA to ensure additional deleterious gene mutations were not accumulated during the process of iPSC production (Supplementary Data 3).

To test the propensity of individual clones to form neurons, we produced embryoid bodies (EBs) in neural induction medium for 2 days and then transferred them to Matrigel-coated plates to form neural rosettes (Fig. 1F–H). We performed quantitative PCR (qRT-PCR) with a panel of primers for early neurogenic genes (Supplementary Data 2 and Fig. 1I). Using these morphologic (rosette formation) and molecular (qRT-PCR) criteria, we identified multiple clones for each patient that had neurogenic potential (Supplementary Data 2). To determine if the iPSCs could make retina in 3D organoid cultures, we used the protocol developed by Sasai[22] and compared retinal formation for each iPSC clone to that of the H9 ESC line using qRT-PCR and immunostaining (Fig. 1J–L and Supplementary Data 2). Overall, we were able to isolate multipotent iPSC clones from each donor with normal karyotype and neurogenic/retinal differentiation competence that retained their germline *RB1* alterations.

**Optimization of retinal specification in 3D organoid cultures.** The original Sasai method for producing 3D retinal organoids was developed and optimized using a highly neurogenic H9 human ESC line[23] and has not been systematically tested for patient-derived iPSCs. While each of our iPSC lines was able to produce retinal organoids using the Sasai method, the efficiency varied from 4% (4/96 for SJRB-iPSC-12) to 27% (26/96 for SJRB-iPSC-7) across lines (Fig. 2A). In order to improve the retinal specification, we carried out several rounds of optimization and made six changes to the procedure. In our modified 3D-RET protocol (Fig. 2A), we added the BMP signaling inhibitor

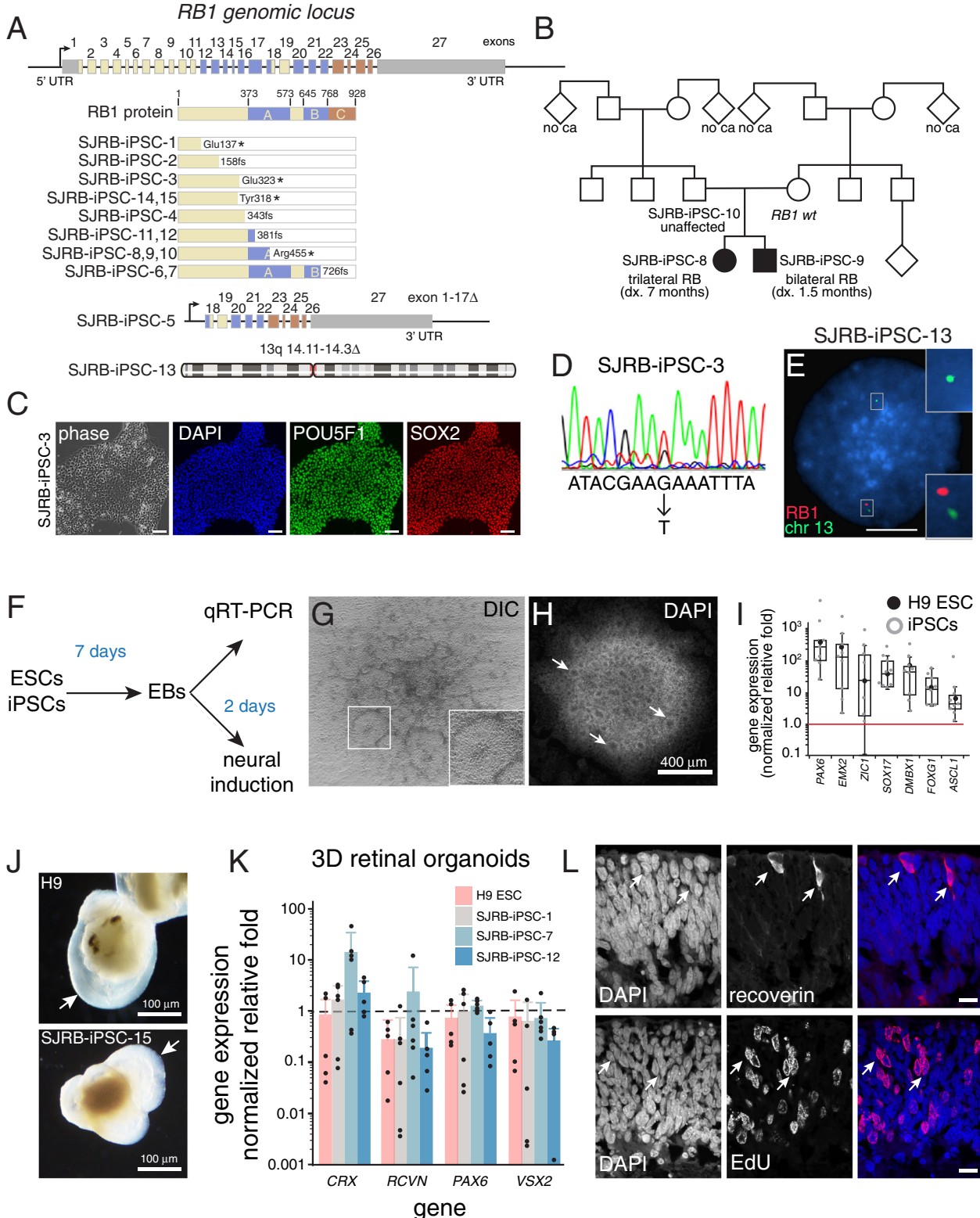

(dorsomorphin) and ROCK inhibitor to promote the neuroectodermal lineage[24–26]. We increased Matrigel to 2% to increase eye-field specification[23]. We change the medium every 2 days rather than every 5 days during the early stages of the eye-field specification to prevent depletion of growth factors. We moved the addition of smoothened agonist (SAG) earlier to day 12 based on a previous publication[27]. We replaced retinoic acid with the more stable synthetic agonist EC23 (Stemgent Inc). Our modified protocol called 3D-RET achieved 22–30% retinal organoid formation across all of our iPSC lines, and the individual organoids from each line were more homogenous (Fig. 2B, C and Supplementary Fig. 2). Using the H9 ESC line and the iPSC lines produced from participants with germline *RB1* alterations, we showed that molecular, cellular, functional, and anatomic features of retinal organoids produced by the Sasai method and the 3D-RET method were indistinguishable in side-by-side comparisons

**Fig. 1 Generation of iPSC lines from patients with germline *RB1* mutations. A** Map of the genomic locus showing each of the 27 exons of the *RB1* gene and the color-coded protein-coding domains. The germline mutations in each patient/family are shown below the full-length protein. **B** Representative pedigree for an unaffected carrier (SJRB-iPSC-10) and two affected children (SJ-iPSC-8 and SJ-iPSC-9). **C** Representative micrographs of SJRB-iPSC-3 colony immunostained for POU5F1 (green) and SOX2 (red) with DAPI (blue) nuclear stain. Staining was repeated on another iPSC line with the same results. **D** Sanger sequencing chromatogram showing the heterozygous nonsense mutation in the *RB1* gene (GAA→UAA). **E** Representative two-color fluorescence in situ hybridization (FISH) of SJRB-iPSC-13 showing the 13q deletion (200 cells were analyzed). **F** Drawing of the neural rosette differentiation experiment. **G**, **H** Differential interference contrast (DIC) and DAPI-stained colony from the neural rosette induction procedure. Arrows and the enlarged box indicate neural rosettes. Three clones of each line were differentiated and each line was able to produce rosettes. **I** Boxplot of the normalized relative fold of neurogenic genes for each iPSC line (n = 15) and H9 ESCs from qRT-PCR of the neural induction assay. **J** Micrograph of representative retinal organoid from H9 ESCs and SJRB-iPSC-15 indicating retina (arrow). **K** GAPDH normalized relative fold gene expression for 3D retinal organoids from representative iPSC lines using H9 ESCs as controls. Each dot represents the mean of two technical replicates for an individual organoid. **L** Micrograph of cryosection of day 45 retinal organoid that was labeled with EdU for 1 h prior to harvest showing recoverin expressing photoreceptors and EdU+ retinal progenitor cells (red) with DAPI counterstain. Each of the 15 lines was sectioned showing similar results in neural retina regions. Box and whisker plots include center line as median, box as Q1 and Q3, and whiskers as 1.5× interquartile range. Scale bars: **C**, 50 μm; **E**, **L**, 10 μm.

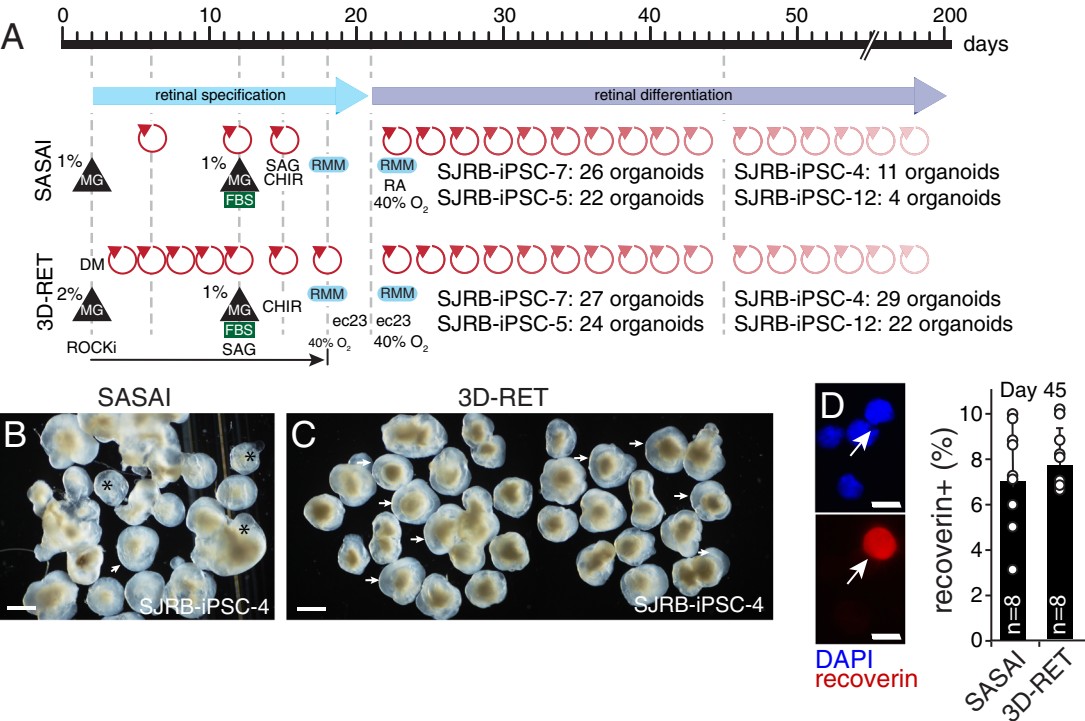

**Fig. 2 The 3D-RET protocol for retinal organoid formation. A** Drawing of the steps in the Sasai and 3D-RET retinal organoid protocol. The number of retinal organoids produced from a 96-well dish is indicated for representative lines for each protocol. Red arrows indicate media changes. **B**, **C** Micrograph of representative retinal organoids using the Sasai and 3D-RET protocols in a side-by-side comparison for SJRB-iPSC-4. All organoids from a 96-well dish were analyzed. Arrows indicate retina organoids and (*) indicates cystic structures that are common in the Sasai protocol. **D** Micrographs of dissociated cell immunofluorescence of retinal organoids and a dot plot showing the percentage of recoverin immunopositive cells from individual retinal organoids (n = 8). MG Matrigel, SAG smoothened agonist, RMM retinal maturation medium, FBS fetal bovine serum, CHIR GSK3 inhibitor, ec23 retinoic acid analog. Scale bars: **B**, **C**, 100 μm. **D**, 5 μm.

(Fig. 2D and Supplementary Fig. 2). While the quality of retinal tissue was similar in each method, the increased efficiency of retinal organoid formation for iPSC lines and the more uniform and consistent size and shape of the organoids, led us to use the 3D-RET protocol for all subsequent experiments.

**Retinoblastoma from human retinal organoids.** Four of the 15 participants in our cohort had surgical enucleation as part of their treatment. We were able to produce O-PDXs from two of those tumors (SJRB-158 and SJRB-124) (Supplementary Data 1). These four patient samples and two O-PDXs will complement our large collection of reference retinoblastoma samples (https://www.stjude.org/CSTN/)[21]. To determine if retinoblastoma can form from 3D

retinal organoid cultures of the patient-derived stem cells, we grew retinal organoids from iPSCs from each patient to day 45 when the tissue was dissociated and injected into the vitreous of immuno-compromised mice (Fig. 3A). As a negative control, we used *RB1* wild-type stem cell (H9 ESCs and GM23710 iPSCs)-derived retinal organoids and as a positive control, we induced *RB1* mutations in exon 4 with CRISPR-Cas9 in all 15 participant derived iPSC lines and H9 cells, hereafter referred to as SJRB-iPSC-1CR - 15CR (Fig. 3B). The CRISPR-Cas9 *RB1* inactivation was intentionally left mosaic (<10% of cells) in the starting stem cell populations to more closely mimic the clonal heterogeneity in human retinoblastoma (Fig. 3C and Supplemental Information). We reverted the germline *RB1* mutation in two of the lines (SJRB-iPSC-4-REV and SJRB-

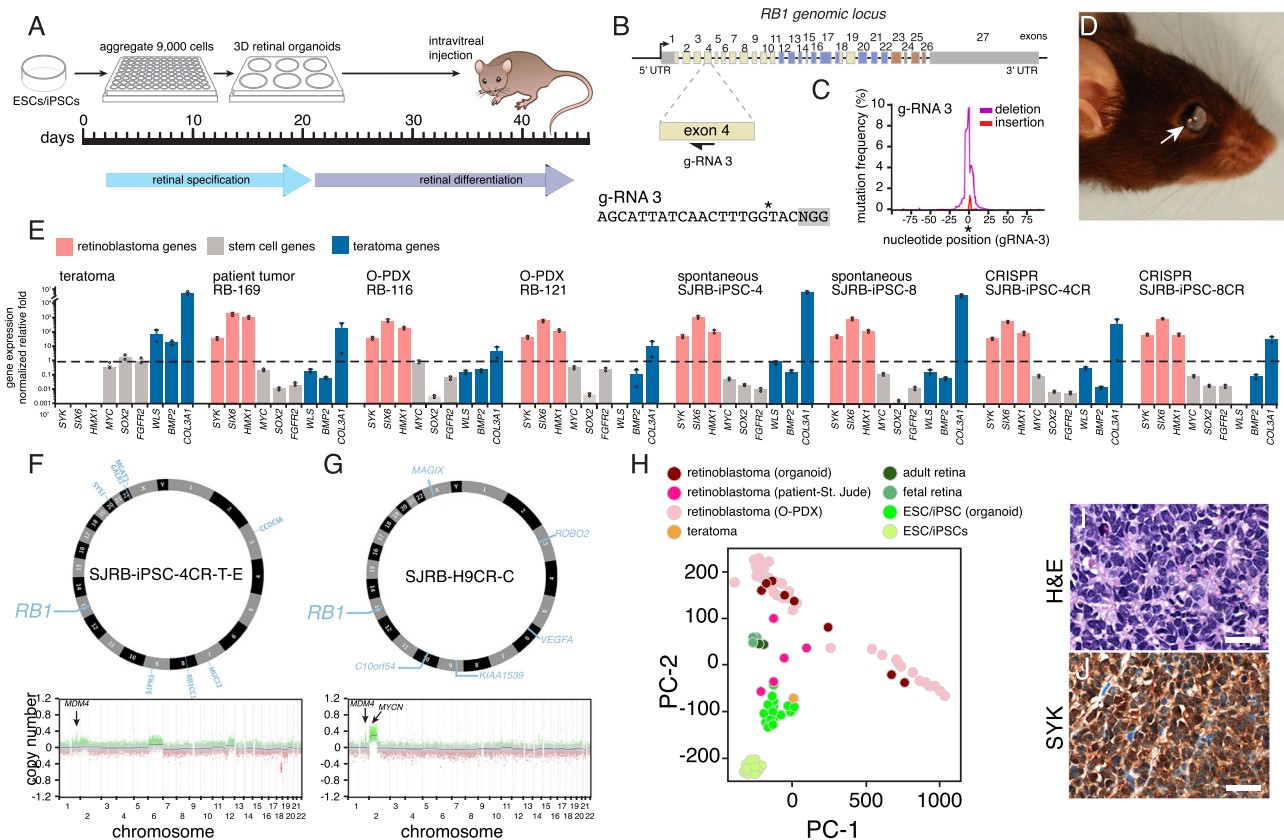

**Fig. 3 Retinoblastoma from 3D retinal organoids. A** Schematic drawing of the retinoblastoma workflow. After 45 days of differentiation, retinal organoids are dissociated and injected into the eyes of immunocompromised mice and they are held for 1 year to wait for tumor formation. **B** Drawing of the *RB1* genomic locus with the location of the gRNA targeting exon 4 and the corresponding sequence. **C** Plot of mutation frequency in a representative iPSC line for gRNA-3 with insertions (red) and deletion (blue) flanking the cut site (*). **D** Photograph of a mouse with retinoblastoma from a retinal organoid. **E** Barplot of qRT-PCR for genes found in the retinoblastoma (*SYK, SIX3, HMX1*), human pluripotent stem cells (*MYC, SOX2, FGFR2*), and teratomas (*WLS, BMP2, COLA1*) from one teratoma, one patient tumor (RB-169), two PDX tumors (RB116 and RB121), two spontaneous retinal organoids-derived tumors (SJRB-iPSC-4 and SJRB-iPSC-8), and two CRISPR-modified retinal organoids (SJRB-iPSC-4CR, SJRB-iPSC-8CR). Each dot is the mean of technical duplicates, the bar is the mean and standard deviation between replicates. All data are normalized to *GAPDH* and plotted relative to H9 ESCs (dashed line). **F, G** Circos plot of representative organoid-derived retinoblastoma (SJRB-iPSC-4CR-T-E and SJRB-H9CR-T-C) showing somatic mutations acquired in the tumor relative to the iPSC/ESC line. The copy number changes across the genome are shown below each tumor. **H** Principal component analysis (PCA) of RNA-seq of organoid-derived retinoblastomas, O-PDXs, patient tumors, iPSC/ESCs, and retinal organoids. **I, J** Hematoxylin and eosin (H&E)-stained organoid-derived tumor showing rosettes and IHC for SYK (brown) which is not present in the normal retina but is upregulated in retinoblastoma. Staining was completed on three tumors with similar results. Scale bars: 25 μm.

iPSC-6-REV) using CRISPR-Cas9 and performed the same tumor formation experiments in parallel with those lines (Supplementary Data 4 and Supplemental Information). We also injected undifferentiated H9 ESCs to form teratomas in the eye as an additional negative control. Each iPSC line and controls were tested with and without the *RB1* CRISPR-Cas9 by two different technicians with three replicate injections per line for a total of more than 500 individual eyes injected (Supplemental Information). After 1 year, we identified 13 independent tumors from iPSC lines SJRB-iPSC-1,2,4,5,6,8 with CRISPR-Cas9 inactivation of *RB1* and 7 independent tumors from SJRB-iPSC-3,8,15 lines without CRISPR-Cas9 inactivation of *RB1* (Fig. 3D and Supplementary Data 4). For H9 ESC retinal organoids, four independent tumors were formed with CRISPR-Cas9 inactivation of *RB1* and none was formed without *RB1* inactivation (Supplementary Data 4). The two lines that had the germline *RB1* mutation reverted did not form tumors after more than a year. We developed a custom Taqman qRT-PCR microfluidic card to rapidly distinguish between teratomas, retinoblastoma, and other malignancies (Fig. 3E and Supplemental Information). None of the tumors that formed from intravitreal

injections of day 45 retinal organoids were teratomas based on qRT-PCR (Supplementary Data 4). Two tumors (SJ-iPSC-15-T-A/ B) that arose rapidly (37 and 100 days) had a pan-neuronal gene expression pattern but were not retinoblastoma based on gene expression (Supplementary Data 4). All other tumors from the patient-derived iPSC retinal organoids (iPSC-RBs) were retinoblastomas that were indistinguishable from patient retinoblastomas or orthotopic patient-derived xenografts (O-PDXs) (Fig. 3E and Supplementary Data 4).

Tumors were propagated and cryopreserved as done previously[6,21,28]. Next, we performed whole-genome sequencing and RNA-seq of each independent iPSC-RB when sufficient tissue was available after initial propagation and screening. The patient-derived iPSC-RBs showed inactivation of the 2nd allele of the *RB1* gene and no other mutations in known cancer genes (Fig. 3F, G, Supplementary Data 4, and Supplementary Fig. 3). Importantly, there were also copy number gains in *MDM4* and *MYCN* which are common in retinoblastomas (Fig. 3F, G, Supplementary Data 4, and Supplementary Fig. 3). In addition, the RNA-seq from iPSC-RBs most closely matched retinoblastomas in principal component

analysis (Fig. 3H and Supplementary Data 4). The difference in O-PDX and organoid-derived retinoblastomas are primarily due to differences in the tumor microenvironment. Vascular endothelial cells, macrophages, and other normal human cells present in the patient tumors are murine in O-PDX and organoid tumors and therefore filtered out in RNA-sequencing analysis. One of the hallmarks of retinoblastoma is the epigenetic deregulation of the *SYK* oncogene which is required for tumorigenesis[6]. SYK RNA and protein were upregulated in iPSC-RBs as found in patient tumors (Fig. 3E, I, J and Supplementary Data 4). Taken together, these data suggest that spontaneous retinoblastoma can form from patient-derived iPSCs differentiated into retinal organoids or be induced by CRISPR-Cas9 targeting of the *RB1* locus in both hESCs and patient-derived iPSCs.

**iPSC-RBs recapitulate the epigenetic and clonal features of retinoblastoma.** High-density DNA methylation arrays (Illumina Infinium 850 K) have been extensively used to classify tumors based on their genome-wide methylation signatures and copy number variations[29–31]. The assay is robust even with small amounts of formalin-fixed paraffin-embedded (FFPE) material. This has been particularly useful for pediatric tumors of the central nervous system[29]. To establish a baseline for retinoblastoma, we profiled 53 retinoblastoma patient tumors, including tumors with histopathological features of differentiation (rosette formation) as well as less differentiated tumors (Fig. 4A, B). Unsupervised hierarchical clustering of the patient tumors revealed that they separate based on differentiation evaluated from the histopathology and DNA copy number variation from the Infinium 850 K array (Fig. 4C). Indeed, the more differentiated tumors had fewer copy number alterations than those with more aggressive undifferentiated histopathologic features and some patient tumors were heterogeneous between regions of differentiated and undifferentiated tumor cells (Fig. 4D–F).

Next, we performed the same DNA methylation array profiling on our collection of iPSC/ESC-derived retinal organoids, retinoblastoma O-PDXs, and organoid-derived retinoblastomas described above. Four of our organoid-derived tumors had sufficient DNA for methylation array profiling and passed our quality control metrics (Supplemental Information). The organoid-derived retinoblastomas clustered more closely with the undifferentiated patient tumors using the same unsupervised methods (Fig. 4G and Supplemental Information). Based on copy number variants, the organoid-derived tumors were intermediate between the differentiated and undifferentiated patient retinoblastomas (Fig. 4G). In unsupervised methylation analysis, the O-PDXs and iPSC-RBs overlapped with retinoblastomas/pineal tumors from a reference collection of 2901 brain tumor methylation profiles[29] and they were clearly separated from the normal retinal organoids and iPSCs in tSNE plots (Fig. 4H–K and Supplementary Data 5).

To further validate the identity and heterogeneity of the retinoblastomas derived from iPSCs, we performed single-cell RNA sequencing on 11 retinoblastoma O-PDXs in biological duplicate for a total of (114,167 cells), 5 healthy adult human retina (24,445 cells), 6 patient retinoblastomas (27,825 cells), and 2 organoid-derived tumors (13,864 cells). We also included human retinal progenitor cells from a publicly available scRNA-seq dataset on human fetal retina[32]. We combined the five healthy adult retina and proliferating fetal retinal progenitor cells and created a reference dataset for all retinal cell types (progenitors, rods, cones, ganglion, horizontal, amacrine, bipolar cells, and Müller glia) as well as non-retinal cell types (vascular endothelial cells and immune cells) using Seurat (v3)[33] (Fig. 5A). In the retinoblastoma samples, the tumor cells were distinguished

from normal cells based on inferred copy number alterations (Supplementary Fig. 4A and Supplemental Information). For each tumor sample, after pairs of cell correspondences between the reference and tumor dataset (anchors) were identified, the cell-type classification was projected and transferred onto each tumor dataset to determine if there were cells with expression profiles similar to specific retinal cell types in the tumors (Fig. 5B). Importantly, the cell cycle genes were removed before the label transfer to prevent bias toward proliferating retinal progenitor cells. Despite excluding the cell cycle genes, the most common cell identity from the label transfer was retinal progenitor cells at 52% (81,283/156,244) followed by rod photoreceptors at 31% (48,591/156,244) (Supplementary Data 6). Indeed, for every patient tumor, O-PDX, and organoid-derived tumor, the most common cell identity was retinal progenitor cells (range 39–63%) and they were enriched in cell cycle genes even though the cell cycle genes themselves were not used for the label transfer (Fig. 5C, D and Supplementary Data 6). To determine if there was evidence of tumor cells with progenitor signatures giving rise to more differentiated tumor cells with photoreceptor gene expression signatures, we performed RNA velocity analysis (Supplemental Information). While some tumors showed evidence of the transition from progenitors to photoreceptors, others showed the opposite pattern (Supplementary Fig. 4). A more definitive clonal analysis will be required to determine the relationship between the cell populations in retinoblastoma.

Previous retinoblastoma single-cell gene expression array analysis of a single O-PDX suggested that tumor cells may have a hybrid gene expression signature of multiple cell types[14]. Consistent with those data, we found that genes that are normally expressed in a mutually exclusive pattern in the normal retina such as *HES6* and *AIPL1* are co-expressed in retinoblastoma tumor cells (Fig. 5E–H). This was also true for cone, rod, and amacrine genes (Fig. 5I and Supplementary Fig. 4E, F). Therefore, individual retinoblastoma tumor cells express a hybrid gene expression signature that does not normally occur during retinal development. Taken together, our scRNA-seq analysis showed that our organoid-derived tumors are indistinguishable from the O-PDXs and the patient tumors in terms of cell identity and proliferation (Fig. 5G and Supplementary Data 6) and all data are available in a Cloud-based viewer (https://pecan.stjude.cloud/static/rbsinglecell).

**Discussion**
We have developed iPSCs from 15 participants with germline *RB1* alterations and we have optimized a 3D retinal organoid culture system for producing human retinoblastoma in the laboratory. We also developed the tools to induce *RB1* mutations in a wild-type human stem cell and produce retinoblastomas indistinguishable from those of patient-derived iPSCs. The organoid-derived retinoblastomas have molecular, cellular, histopathologic, genetic, epigenetic, and clonal features that are indistinguishable from patient tumors and O-PDX models. In contrast with O-PDXs, this model is not reliant on patient tumor tissue, which in the case of retinoblastoma is usually only available after enucleation. In addition, our model can be used to derive multiple tumors from the same patient and can be used to generate tumors from carriers who never developed retinoblastoma. The process of producing retinoblastoma in our system was inefficient (<5%) and time-consuming (12–18 months per tumor for engraftment and propagation). Still, this is only slightly longer than time to engraftment of a retinoblastoma O-PDX. Subsequent rounds of differentiation and injection after screening organoids for high percentages of retinal tissue have greatly increased efficiency, including the formation of at least one tumor from all CRISPR-mutated SJRB-iPSC lines that were

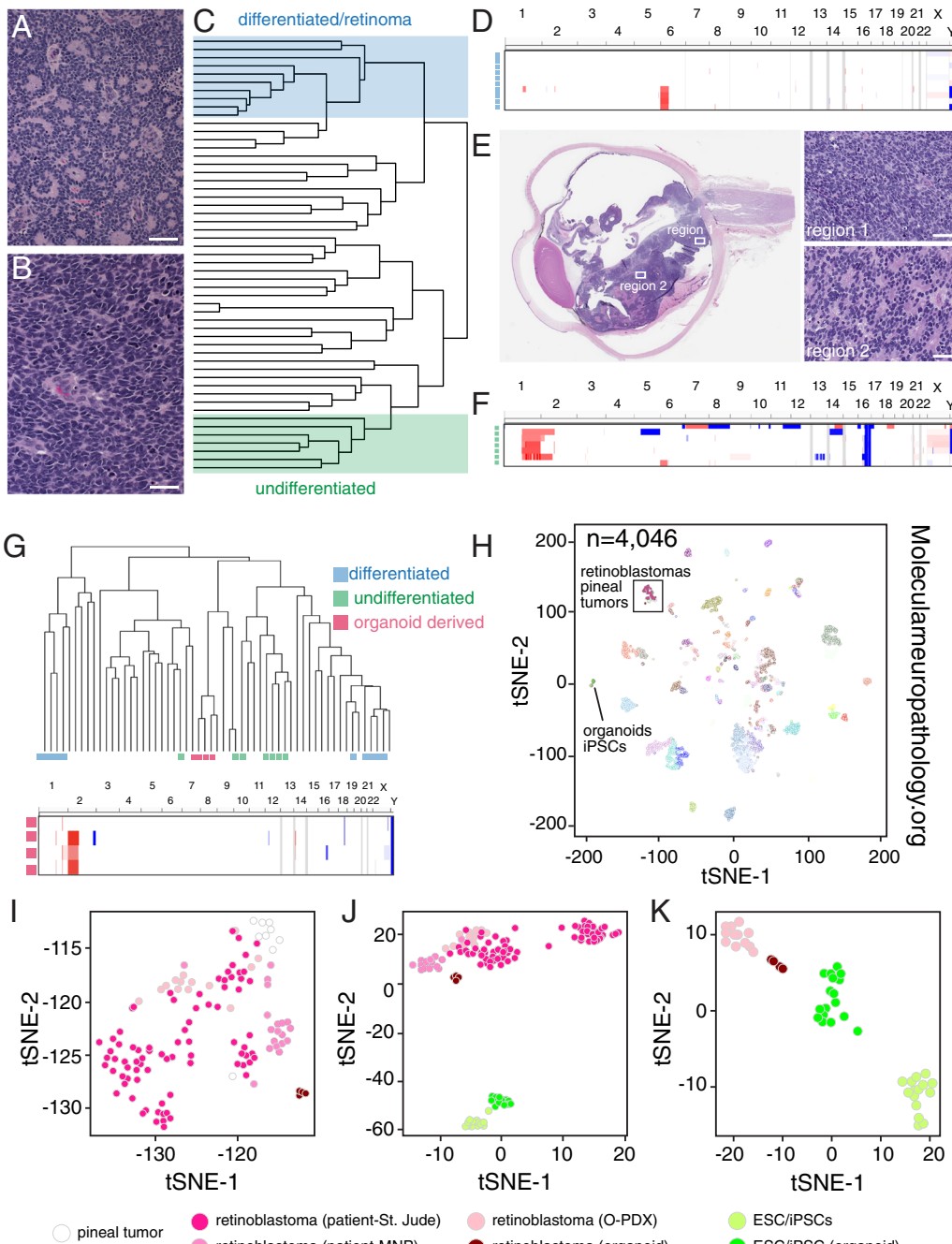

**Fig. 4 DNA methylation profiling of retinoblastoma. A**, **B** Hematoxylin and eosin (H&E) staining of differentiated and undifferentiated patient retinoblastoma. **C** Unsupervised hierarchical clustering of retinoblastoma tumors showing separation of the differentiated and undifferentiated samples. **D**–**F** Copy number variation in the differentiated retinoblastomas (**D**) and undifferentiated patient retinoblastomas (**F**) and H&E staining (**E**) of a patient eye that has regions of both undifferentiated (region 1) and differentiated (region 2) tumor, suggesting intratumor cellular heterogeneity. **G** Copy number variation and clustering of the retinal organoid-derived tumors. **H** t-distributed stochastic neighbor embedding (tSNE) plot of the St. Jude retinoblastomas, retinal organoid-derived tumors, retinal organoids, and iPSC/ESCs integrated with the Molecular Neuropathology (MPN) database of brain regions and pediatric brain tumors. **I** tSNE plot of the boxed region in (**H**) showing clustering of the organoid-derived tumors with patient retinoblastomas. **J** tSNE plot of the patient, O-PDX, and organoid-derived retinoblastomas with iPSC/ESCs and normal retinal organoids. **K** tSNE plot of O-PDX and organoid-derived retinoblastomas with iPSC/ESCs and normal retinal organoids. Scale bars: 25 μm.

selected for injection. The introduction of additional perturbations such as ectopic expression of MDM2/4, MYCN, or SYK may accelerate tumorigenesis. Previous attempts to generate retinoblastoma from genetically modified H9 ESCs may have failed because they did not allow enough time for tumors to grow (60–90 days) or the lack of normal retinal development in the absence of *RB1*[34]. Our approach using a mosaic approach with the same H9 ESC produced multiple independent tumors in 200–300 days. None of our tumors was teratomas, indicating that retinal organoid differentiation prior to intravitreal injection was sufficient to eliminate any residual stem cells. However, we did identify two tumors using this method that had neuronal features but were not retinoblastomas. Therefore, it is essential to implement robust diagnostics for retinoblastoma from retinal

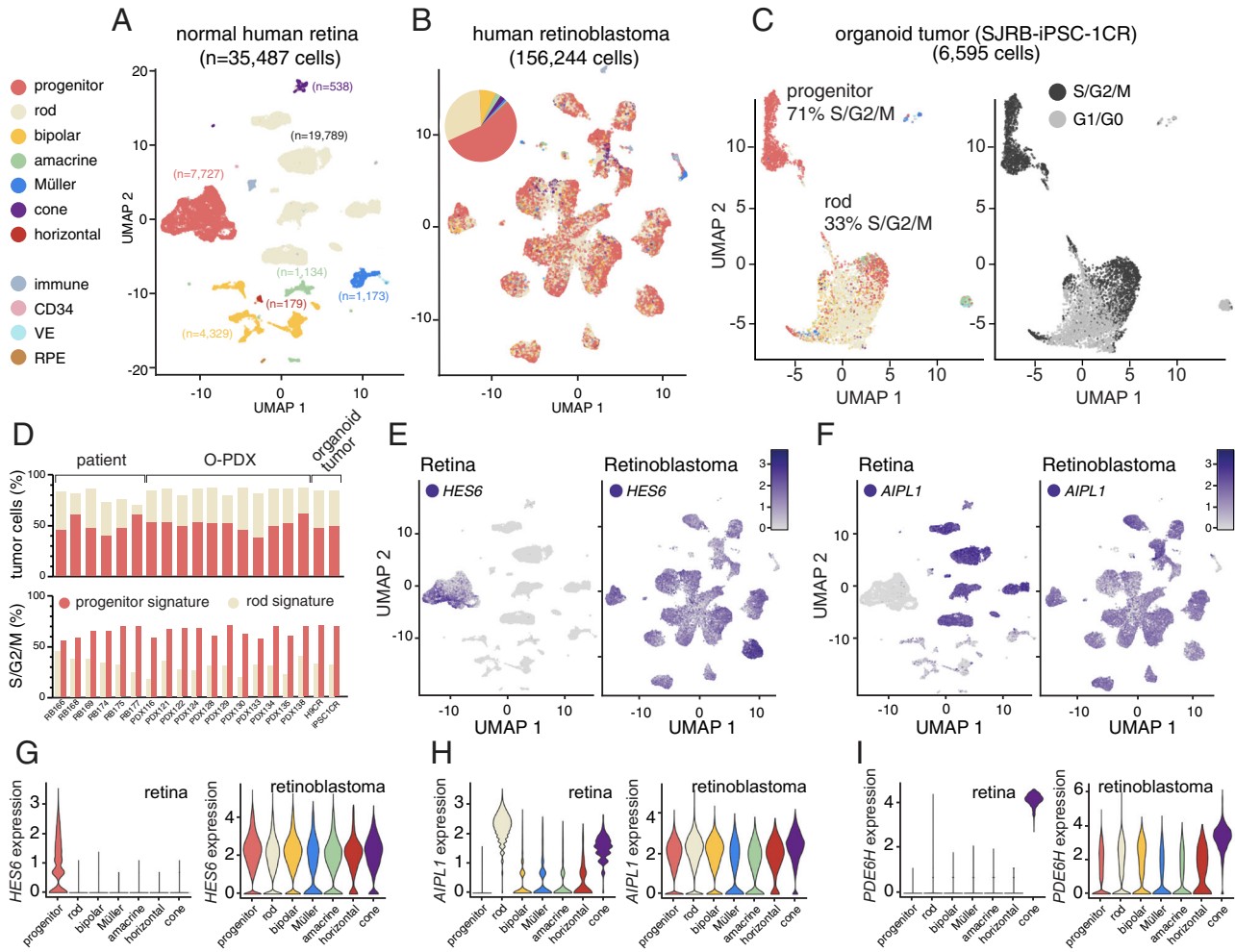

**Fig. 5 Cellular heterogeneity of retinoblastomas. A** Uniform manifold approximation projection (UMAP) plot of scRNA-seq of normal human retinal cells including retinal progenitor cells and all adult retinal cell types. **B** UMAP plot of scRNA-seq of human retinoblastoma from patient tumors, O-PDXs, and organoid-derived tumors. The label transfer for cell identity is displayed, and the overall numbers are represented in the piechart in the upper left corner. **C** Representative UMAP plot of one of the organoid-derived tumors showing cells with the progenitor cell signature and rod signature and the relative distribution of gene expression profile for proliferation. **D** Barplot of the proportion of single cells with the retinal progenitor cell gene expression signature and the rod signature for each tumor (upper panel) and the proportion of those cells that are proliferating (S/G2/M, lower panel) for each tumor analyzed by single-cell RNA sequencing (n = 19). **E, F** UMAP plot of gene expression for normal retina and retinoblastoma for a representative retinal progenitor cell gene (*HES6*) and a photoreceptor gene (*AIPL1*). **G, H, I** Violin plot of the distribution of expression of a progenitor gene (*HES6*), a photoreceptor gene (*AIPL1*) and a cone gene (*PDE6H*) across retinal cell types and retinoblastomas. The colors match the colors in (**A, B**).

organoids that include molecular, cellular, genetic, and epigenetic features. This study also provided new insights into the cellular identity within retinoblastoma demonstrating a bias toward retinal progenitor cells and rods. It is possible that the highly proliferative tumor cells with the retinal progenitor cell identity give rise to more differentiated tumor cells that have features rods and other neurons but lineage tracing will be required to test that hypothesis. This complex process of the tumor cells progressing through some aspects of retinogenesis may have confounded previous attempts to identify the retinoblastoma cell of origin from gene expression analysis of bulk tumors. Indeed, scRNA-seq shows a hybrid cellular phenotype, which may simply reflect the multipotency of retinal progenitor cells. Our retinal organoid system will be an important tool for determining if those tumor cell populations are distinct clones or if they represent the dynamic changes in gene expression of individual clones over time. The tumor modeling described here may also be useful for testing novel therapeutic combinations for individual patients.

## Methods

**Patients**. RETCELL (NCT02193724), a protocol to establish the feasibility, validation, and differentiation of induced pluripotent stem cells produced from patients with heritable retinoblastoma, was approved by the St. Jude Children's Research Hospital Institutional Review Board and open to accrual in July 2014. Written informed consent was obtained from each participant or participant's parent/guardian. Eligibility criteria included a family history of retinoblastoma with an identified germline *RB1* mutation, diagnosis of bilateral retinoblastoma, or diagnosis of unilateral retinoblastoma with germline *RB1* mutation, *MYCN* amplification, or 13q deletion identified. Samples obtained from participants included skin biopsy or collection of peripheral blood mononuclear cells. Skin biopsies were performed while the patient was under sedation for clinical purposes (e.g., exam under anesthesia). Peripheral blood samples were obtained at any time the participant was scheduled for routine labs; if the participant was not an active patient, the blood draw occurred as a separate, research-only blood draw.

Specimens were de-identified after collection in the St. Jude Biorepository and shipped overnight to the Waisman Center for further processing. One sample required a repeat blood draw due to difficulties encountered with the shipping process.

A total of 11 patients were enrolled (4 female), with 4 parental samples obtained simultaneously due to positive family history. Four samples were obtained by skin biopsy at the time of anesthesia for examination of the eyes as part of routine care, and 11 samples were obtained with the peripheral blood draw. Germline blood samples were obtained from 14 of 15 participants; one participant is not available

for germline testing at this time. Participants were selected and approached for inclusion in this study based on their clinical presentation to represent a broad spectrum of penetrance in heritable retinoblastoma disease: four with no family history diagnosed at <12 months of age, two with 13q deletion (manifested as a unilateral and bilateral disease), and four family cohorts ($n = 9$), including two families where the parent was previously unknown to carry the *RB1* mutation until the disease was detected in the offspring.

All participants are alive, with no participant currently receiving treatment for active disease. One participant, treated for retinoblastoma as an infant, had a second malignancy noted 12 years after diagnosis; no other cases of second malignancy are reported. Visual acuity is routinely followed in retinoblastoma survivors when patients are old enough to comply with standard visual acuity measurements. Visual acuity is not available for two participants who were previously unknown to be affected by retinoblastoma and one patient who is not in active treatment for the disease; all participants are living as independent adults. Six participants are too young or are noncompliant with standard visual acuity measurements for formal visual testing. In the remaining six participants, five have 20/20 vision in at least one eye (the least affected or unaffected eye at diagnosis), and one patient has preserved light perception.

**Fibroblast reprogramming.** Fibroblasts were derived from a skin biopsy and cultured in fibroblast medium: DMEM supplemented with 10% FBS (all Life Technologies). In total, $1 \times 10^6$ fibroblasts were reprogrammed using CytoTune Sendai reprogramming Kit (Life Technologies A16518), transduced with the recommended multiplicity of infection (MOI) for each vector (KOS MOI = 5, hc-Myc MOI = 5, hKlf4 MOI = 3) for 24 h and seeded on mouse embryonic fibroblast (MEF) feeder, switched to human embryonic stem cells (hESC) medium 2 days post transduction: DMEM/F12 with GlutaMAX, 10% KnockOut Serum Replacement, 1× nonessential amino acids (Life Technologies), 0.1 mM 2-mercaptoethanol (Sigma), and 10 ng/mL bFGF (Waisman Center). Picked induced pluripotent stem cell (iPSC) clones were cultured on MEF feeder/hESC medium first and subsequently adapted in feeder-free conditions on matrigel (Corning) in the presence of E8 medium (Life Technologies). iPSC lines were routinely passaged using 0.5 mM EDTA and thawed in the presence of ROCK inhibitor (Y-27632, Sigma) 24 h after thawing.

**Blood cell reprogramming.** Peripheral blood mononuclear cells (PBMCs) were isolated from a remotely collected blood sample using a Vacutainer CPT cell preparation tube with sodium citrate (BD Biosciences), separated by centrifugation, and cultured in PBMC medium: RPMI-1640 plus 10% FBS, GlutaMAX and sodium pyruvate (all Life Technologies). In all, $1 \times 10^6$ PBMCs were reprogrammed using CytoTune Sendai reprogramming Kit (Life Technologies A16518), transduced with the recommended MOI for each vector (KOS MOI = 5, hc-Myc MOI = 5, hKlf4 MOI = 3) for 24 h, and seeded on MEF feeder, 2 days post transduction switched to hESC medium. hESC medium: DMEM/F12 with GlutaMAX (Life Technologies), 10% KnockOut Serum Replacement (Life Technologies), 0.1 mM 2-mercaptoethanol (Life Technologies), 1× nonessential amino acids (Life Technologies), and 10 ng/mL bFGF. Picked iPSC clones were cultured on MEF/hESC medium first and subsequently adapted and cultured in a feeder-free condition on matrigel (Corning) in the presence of E8 medium (Life Technologies). iPSC lines were routinely passaged using 0.5 mM EDTA and thawed in the presence of ROCK inhibitor (Y-27632, Sigma) 24 h after thawing.

**Immunostaining iPSCs.** iPSCs were cultured on a chamber slide and were fixed in 4% PFA overnight at 4 °C, washed (three times with PBS), and blocked in blocking solution for 1 h at room temperature (RT). Primary antibody diluted in blocking solution was added and incubated overnight (O/N) at 4 °C. Cells were stained with the following antibodies: Oct3/4 (1/1000 dilution Santa Cruz, SC-5279), Sox2 (1/1000 dilution R&D AF2018), Nanog (1/200 dilution Stemgent 09-0020). Slides were washed, and secondary antibody diluted in blocking solution was added and incubated 1 h at RT. Slides were washed, incubated with 4,6-diamidino-2-phenylindole (DAPI, LifeTech D1306) solution for 10 min at RT, washed, and mounted using Prolong Gold (ThermoFisher P10144). Images were captured using either a Nikon C1 laser scanning microscope or a Nikon Eclipse E600 microscope.

**Culturing iPSCs.** ESCs (embryonic stem cells) and iPSCs were cultured and maintained in mTESR (StemCell Tech #85850) on plates coated with matrigel (Corning #354277) at 37 °C, 5% CO$_2$, and 5% O$_2$. Media was changed daily. Cells were passaged at 80% confluency by adding 1 mL Versene (Gibco #15040-066) and incubating 5 min at RT. The Versene was aspirated off, and 1 mL mTESR was used to wash the cells off the bottom of the well. The cells were then split 1:6 into new matrigel-coated wells.

**Validation of mutations.** DNA was extracted from iPSC lines using the DNeasy blood and tissue kit (Qiagen #65904). PCR was performed using primers (Table S1) based on which exon was mutated in the germline sample.

SJRB-iPSC13 8.7 Mb deletion was confirmed by FISH (fluorescent in situ hybridization) and SJRB-iPSC 5 1-17 exon deletion was confirmed by WGS (whole-genome sequencing).

**Fluorescence in situ hybridization.** Two fosmid clones (WI2-1468C23 and WI2-0655F03) that localize to the region that is expected to be deleted were combined with an internal control probe (Pan3) located 20 megabases centromeric to the RB1 5′ locus. The two RB1 5′ fosmid DNAs were labeled with a red-dUTP (Alexa Fluor 594, Molecular Probes) by nick translation. The labeled RB1 5′ probe and a green-dUTP (Alexa Fluor 488, Molecular Probes) labeled chromosome 13 control probe (RP11-904N23/13q12.2) were combined with sheared human cot DNA and hybridized to interphase nuclei and metaphase chromosomes derived from the cell line by routine cytogenetic methods in a solution containing 50% formamide, 10% dextran sulfate, and 2× SSC. The chromosomes were then stained with DAPI (LifeTech D1306) and analyzed.

**Retinal organoid immunostaining.** Organoids were fixed in 4% PFA overnight at 4 °C, washed 3× with PBS, incubated in 30% sucrose at RT until the organoids were saturated and sunk to the bottom of the solution. Then the organoids were embedded in OCT (Scigen #4583). Cryosections were fixed with 4% PFA for 1 h at RT, washed (three times with PBS), and blocked in blocking solution for 1 h at RT. Primary antibody diluted in blocking solution (Table S2) was added and incubated O/N at 4 °C. Slides were washed, and secondary antibody diluted in blocking solution was added and incubated 1 h at RT. Slides were washed and incubated with the ABC kit solution (Vector Labs PK-4000) for 30 min at RT, washed, incubated in Tyramid solution for 10 min at RT, washed, incubated with DAPI solution for 10 min at RT, washed, and mounted using Prolong Gold (P10144, ThermoFisher).

**Karyotyping.** Cells were treated with nocodazole (0.1 µg/mL) for a 16 h incubation at 37 °C. The nocodazole media was aspirated. In total, 1–2 mL of Accutase (Stem Cell Technologies #7920) was added to the wells, incubated for 4 min at 37 °C then 2 mL of RPMI-1640 complete media was added to the wells and pipetted five times to get a single-cell suspension. The suspension was transferred to a 15-mL centrifuge tube and centrifuged at 450 rcf for 5 min. The cell pellet was resuspended in buffered hypotonic solution for 8 min at RT. At the end of the hypotonic incubation, 1–2 drops of cold Carnoy's fixative were added to the centrifuge tube and then centrifuged at 450 rcf for 5 min, the pellet was resuspended in cold 3:1 Carnoy's fixative (3 parts methanol:1 part acetic acid), and incubated for 15 min at RT. The cells were centrifuged at 450 rcf for 5 min, the pellet was resuspended in cold Carnoy's fixative, incubated for 10 min at RT. This step was repeated once more. The cells were spread onto a glass slide using a Pasteur pipette. The slides were allowed to air dry (~2–3 days) for optimal banding of the chromosomes with trypsin and Wright's stain. Images were acquired using a Zeiss Plan-Apochromat ×100 objective on a Zeiss Axio Imager.Z2 microscope and GenASIs scanner (Applied Spectral Imaging BandView software version 8.1).

**SKY.** Cells were treated with nocodazole (0.1 µg/mL) for a 16 h of incubation at 37 °C. The nocodazole media was aspirated. In total, 1–2 mL of Accutase (StemCell Tech #7920 was added to the wells, incubated for 4 min at 37 °C then 2 mL of RPMI-1640 complete media was added to the wells and pipetted five times to get a single-cell suspension. The suspension was transferred to a 15-mL centrifuge tube and centrifuged at 450 rcf for 5 min. The cell pellet was resuspended in buffered hypotonic solution for 8 min at RT. At the end of the hypotonic incubation, 1–2 drops of cold Carnoy's fixative were added to the centrifuge tube and then centrifuged at 450 rcf for 5 min, resuspended in cold 3:1 Carnoy's fixative (3 parts methanol:1 part acetic acid), and incubated for 15 min at RT. The cells were centrifuged at 450 rcf for 5 min, resuspended in cold Carnoy's fixative, incubated for 10 min at RT. This step was repeated once more. The cells were spread onto a glass slide using a Pasteur pipette. The slides were air-dried at RT. Spectral karyotyping of chromosomes was performed using a human SkyPaint probe (Applied Spectral Imaging, Carlsbad, CA) per the procedure recommended by the manufacturer. Probes were detected using Applied Spectral Imaging's concentrated antibodies detection kit (CAD), as described by the manufacturer. After the detection procedure was completed, the cells were then stained with DAPI (80 µg/mL) and analyzed. Images were acquired with a Nikon Eclipse E600 fluorescence microscope equipped with an interferometer (Spectra Cube: Applied Spectral Imaging) and a custom-designed filter cube (Chroma Technology Corporation, Rockingham, VT). SKY analysis was done using HiSKY version 7.2 software (Applied Spectral Imaging).

**Flow cytometry.** In total, 1 mL of Accutase (StemCell Tech #7920) was added to each well of a six-well plate containing iPSCs at 60–80% confluency. Plates were incubated at 37 °C for 3 min. In total, 2 mL iPSC dilution medium (mTESR + 2 µM T Thiazovivin, StemCell Tech #72254) was added to each well and pipetted 8–10 times to resuspend cells. The solution was passed through a 40-µm cell strainer and centrifuged at 300 rcf, 4 min. iPSCs were resuspended in 2 mL iPSC dilution medium. Cells were aliquoted into ten 15-mL tubes with a final volume of 200 µL. All tubes were centrifuged at 300 rcf for 4 min, at 4 °C. Control cells were resuspended in 200 µL chilled blocking solution (iPSC dilution media + 20% BSA) with or without 4 µL of antibodies PE-conjugated anti-TRA-1-81(BD, #560885) or FITC-conjugated anti-SSEA4 (BD, #560126). Experimental cells were resuspended in 500 µL of chilled blocking solution with 10 µL of each antibody. Cells were

incubated on ice for 30 min in a darkened hood then washed by adding 5 mL of the chilled blocking solution to each tube, centrifuging at 300 rcf, 4 min, at 4 °C. The wash was repeated once. iPSCs were resuspended in iPSC dilution media and left on ice. Cells were analyzed by flow cytometry (FACSDiva 8.0.1), at least 50,000 events were collected.

**Immunocompromised mice.** All animal procedures and protocols were approved by the St. Jude Laboratory Animal Care and Use Committee. All studies conform to federal and local regulatory standards. Female C57BL/6 SCID mice were purchased from Jackson Laboratories (strain code 001913). Mice were housed on ventilated racks on a standard 12-h light–dark cycle.

**Trilineage assays.** iPSC lines were cultured to 40–50% confluency in mTESR in a six-well dish then media was changed to Nutristem media (Biological Industries #05-100-1A). Cells were incubated at 37 °C, 5% $O_2$, 5% $CO_2$, O/N. In total, 1 mL of Accutase was added to the well-containing iPSCs at 60–80% confluency. Plates were incubated at 37 °C for 3 min. In all, 2 mL of plating medium (Nutristem medium + 2 μM Thiazovivin) was added to the well and pipetted 8–10 times to obtain a single-cell suspension. Cells were centrifuged at 300 rcf for 4 min and resuspend in 2 mL plating medium and counted. In total, $1 \times 10^6$ cells were aliquoted, and the volume was brought to 10 mL with a plating medium and pipetted with a p1000 tip to create a homogenous cell suspension. In total, 100 μl of cell suspension was plated per well into one Nunclon Sphera 96 U plate. The plate was spun at 100 rcf for 2 min, the bucket orientation was switched, and the plates were spun again at 100 rcf at 2 min. Plates were incubated at 37 °C, 20% $O_2$, 5% $CO_2$ O/N. Four columns of embryoid bodies (EBs) (32 EBs each) were used for each of the three lineage differentiation protocols outlined below.

*Ectoderm differentiation.* On days 1–4, 50 μL of media was removed and replaced with 50 μL of Neural Induction Media (NIM, StemCell Tech #5835). On day 5, a wide bore p1000 tip was used to collect EBs from all 32 wells. EBs were pipetted onto a 40-μm filter and washed with 1 mL NIM. The filter was then flipped and the EB's were washed off the filter with 3 mL NIM. The EB's were plated 1.5 mL per well onto two Matrigel-coated wells of a 12-well plate. On days 7 and 9, the media was changed with 1.5 mL NIM. On day 11, the cells were harvested by adding 1 mL of Accutase to the well and incubating at 37 °C for 3 min. In all, 2 mL NIM was added, and the cells were centrifuged at 300 rcf for 4 min. The media was removed, and the cells were frozen for preparation for the Taqman Array (see below).

*Mesoderm differentiation.* On days 1 and 2, 50 μL of media was removed and replaced with 50 μL of Mesoderm Day 1 media (GMEM; Sigma #G5154, 20% KSR; ThermoFisher #10828028, 6 μM CHIR; StemCell Tech #72254). On day 3, 50 μL of media was removed and replaced with 50 μL of Mesoderm Day 3 media (GMEM, 20% KSR, 50 ng/μL BMP4; ThermoFisher #PHC9534). On day 5, a wide bore p1000 tip was used to collect EBs from all 32 wells. EBs were pipetted onto a 40-μm filter and washed with 1 mL of the Mesoderm Day 3 media. The filter was then flipped, and the EB's were washed off the filter with 3 mL of the Mesoderm Day 3 media. The EB's were plated 1.5 mL per well onto two Matrigel-coated wells of a 12-well plate. On days 7 and 9, the media was changed with 1.5 mL of Mesoderm Day 3 media. On day 11, the cells were harvested by adding 1 mL of Accutase to the well and incubating at 37 °C for 3 min. In total, 2 mL Mesoderm Day 3 media was added, and the cells were centrifuged at 300 rcf for 4 min. The media was removed, and the cells were frozen for preparation for the Taqman Array (see below).

*Endoderm differentiation.* On days 1 and 2, 50 μL of media was removed and replaced with 50 μL of Endoderm Day 1 media (GMEM, 20% KSR, 6 μM CHIR, 100 ng/mL Activin A; PerproTech #120-14 P, 0.5% FBS). On day 3, 50 μL of media was removed and replaced with 50 μL of Endoderm Day 3 media (GMEM, 20% KSR, 6 μM CHIR, 100 ng/mL Activin A, 1% FBS). On day 5, a wide bore p1000 tip was used to collect EBs from all 32 wells. EBs were pipetted onto a 40-μm filter and washed with 1 mL of the Endoderm Day 5 media (GMEM, 20% KSR, 6 μM CHIR, 100 ng/mL Activin A, 2% FBS). The filter was then flipped, and the EB's were washed off the filter with 3 mL of the Endoderm Day 5 media. The EB's were plated 1.5 mL per well onto two Matrigel-coated wells of a 12-well plate. On days 7 and 9, the media was changed with 1.5 mL of Endoderm Day 5 media. On Day 11, the cells were harvested by adding 1 mL of Accutase to the well and incubating at 37 °C for 3 min. In total, 2 mL Day 3 media was added, and the cells were centrifuged at 300 rcf for 4 min. The media was removed, and the cells were frozen for preparation for the Taqman Array (see below).

**Neural/Rosette assay.** In all, 1 mL of Accutase was added to 2–3 wells of iPSCs at 60–80% confluency. Plates were incubated at 37 °C for 3 min. In total, 2 mL NIM was added to the well and pipetted 8–10 times to obtain a single-cell suspension. Cells were centrifuged at 300 rcf for 4 min and resuspend in 1 mL NIM and counted. In total, $2.7 \times 10^6$ cells were aliquoted and the volume was brought to 1 mL with NIM and pipetted with a p1000 tip to create a homogenous cell suspension. The cell suspension was plated per well into one well of a pre-prepared 24-well Aggrewell 800 plate. The plate was spun at 100 rcf for 2 min, the bucket orientation was switched, and the plates were spun again at 100 rcf at 2 min. An

additional 0.5 mL of NIM was added, and the plates were incubated at 37 °C, 20% $O_2$, 5% $CO_2$ O/N. Each day, 1 mL of NIM removed and 1 mL fresh NIM was added.

*Neuronal gene expression assay.* On day 7, a wide bore p1000 tip was used to collect the EBs. The EBs were pipetted onto a 40-μm filter and washed with 1 mL of the NIM. The filter was then flipped, and the EB's were washed off the filter with 3 mL of the NIM. The EBs were allowed to sink to the bottom of the tube, and the media was removed. for preparation for the Taqman Array (see below).

*Rosette-formation assay.* On day 5, a wide bore p1000 tip was used to collect the EBs. The EBs were pipetted onto a 40-μm filter and washed with 1 mL of the NIM. The filter was then flipped, and the EB's were washed off the filter with 3 mL of the NIM. The EB's were plated 3 mL per well onto one well of a matrigel-coated six-well dish. The media was changed on days 7–11, and rosette formation was observed.

**Retinal differentiation.** In all, 1 mL of Accutase was added to one well of a six-well plate containing iPSCs at 60–80% confluency. Plates were incubated at 37 °C for 3 min. In total, 2 mL iPSC dilution medium (mTESR + 2 μM ROCKi Stem Cell Tech #72304) was added to each well and pipetted 8–10 times to resuspend cells and centrifuged at 300 rcf for 4 min. iPSCs were resuspend in 2 mL GMEM Retinal media (GMEM, 20% KSR, 1× NEAA; Gibco #11140-050, 1× sodium pyruvate; Gibco #11360070, 1× ß-mercaptoethanol; Gibco #21985-023, 1× anti anti; Gibco #15240-062, 2 μM ROCKi) supplemented with 3 μM IWRe (Selleck Chem #S7086) and counted. In total, $0.9 \times 10^6$ cells were aliquoted, and the volume was brought to 10 mL with GMEM Retinal media supplemented with 3 μM IWRe and pipetted with a p1000 tip to create a homogenous cell suspension. The cell suspension was plated at 100 μL per well into one 96 U bottom plate. The plate was spun at 100 rcf for 2 min, the bucket orientation was switched, and the plates were spun again at 100 rcf at 2 min. The plates were incubated at 37 °C, 20% $O_2$, 5% $CO_2$ O/N.

On day 2, 30 μL of GMEM Retina supplemented with 3 μM IWRe, 2 μM Dorsomorphin (Tocris #3093), and 2% v/v GFR-Matrigel (Corning #354230) was added to each well. On days 4, 6, 8, and 10, 50 μL per well of media was changed with GMEM Retina media + 2% v/v GFR-Matrigel. On day 12, a wide bore p1000 tip was used to collect the EBs and transfer them to two wells of a six-well low-attachment plate. In all, 2 mL of GMEM Retina media supplemented with 10% v/v ES-FBS, 100 nM SAG, 1% v/v GFR-Matrigel was added. On day 15, the media was replaced with GMEM Retina media supplemented with 10% v/v ES-FBS (Gibco #15240-062), 100 nM SAG (StemCell Tech #73414), 3 μM CHIR (StemCell Tech #72054), 1% v/v GFR-Matrigel. On day 18, the media was replaced with Retinal Maturation media (DMEM:F12 Gibco #11330-032, 1× N2 Gibco #17502-048, 10% ES-FBS, 1× anti anti, 0.5 μM EC23 ReproCell #SRP002) and incubated at 40% $O_2$ (37 °C, 5% $CO_2$). The media was replaced every 3 days until the retina was harvested.

**Retinal differentiation scoring.** The hESC H9 and a selection of iPSC were differentiated using the method above and the original method from Nakano et al.[23] On day 45 of culture, organoids were dissociated individually. Each organoid was placed in an Eppendorf tube with 200 μL of PBS−/− and 20 μL of trypsin (10 mg/mL Sigma T9935) for 10–15 min at 37 °C then 20 μL of soybean trypsin inhibitor (STI 10 mg/mL, Sigma T6522) and 20 μL DNase (Sigma D4513) was added, and the cells were incubated for 5 min at 37 °C. For all cells, a portion of the cells was added to a coated slide and allowed to sit for 1 h before immunostaining. The remaining cells were spun down, and RNA was extracted using Trizol (LifeTech #15596018), cDNA was made using Superscript III (Invitrogen 18080-051), and qPCR (Table S3) was run using SYBR Select Master Mix for CFX (ThermoFisher #4472942) with standard cycling conditions (95 °C for 2 min and 40 cycles of 95 °C for 30 s, 60 °C for 1 min).

**Taqman qRT-PCR.** RNA was extracted from samples using Trizol. cDNA was prepared using a High-Capacity RNA-to-cDNA Kit (ThermoFisher #4387406). Custom TaqMan Arrays were created for neural gene expression assay, the trilineage assay, and to identify retinoblastoma (Table S4).

**Multielectrode array.** For recording evoked response to optical stimulation in retinal organoids, the retinal ganglion cells inside the organoids need to be in contact with the electrodes on the MEA plate. Therefore, retinal organoids, no matter the size, should first be sliced before plating. Slice the retinal organoids using a pair of fine forceps by pinching the organoid below a smooth, translucent, and laminated-looking surface along an axis parallel to the surface of the organoid. DO NOT pinch perpendicular to the surface of the organoid, and DO NOT pinch too far from the laminated surface of an organoid such as through the center of the organoid.

Growth factor reduced Matrigel (Corning) was diluted 1:3 in chilled media containing retinal maturation media and 10% ES-FBS and kept on ice. Each slice was then plated onto one well of a 24-well Cytoview plate (Axion Biosystems), where each well contained a grid of 16 electrodes. After placing the organoid slice onto a dry and empty well using a sterile flat spatula, 15 μL of the chilled and

diluted Matrigel was added onto the slice. A P10 tip was used to position the slice to the center of the well over the grid of electrodes. Three such plates were prepared. The plates were left at 37 °C for 3 h for the Matrigel to solidify over the organoid slices. After 3 h, 500 μL of retinal maturation media freshly supplemented with 0.5 μM EC23 was added slowly to each well. Leave at 37 °C and 40% $O_2$ undisturbed for 3 days. On day 3, replace 300 μL media in each well with 1:1 Retinal maturation media: complete BrPhys (BrPhys basal media + 10% ES-FBS + 100× N2 + 50× B27 w/o Vit.A + 10 ng/mL BDNF + 10 ng/mL GDNF), freshly supplemented with 0.5 μM EC23. On day 6, replace 300 μL media in each well with complete BrPhys, freshly supplemented with 0.5 μM EC23. Henceforth, change media every 3 days.

We used the MaestroEdge (Axion Biosystems) to record evoked activity to optic stimulation. Organoids were illuminated using Lumos that allow stimulation with the following wavelengths: 475 nm (BLUE LED), 612 nm (ORANGE LED), 530 nm (GREEN LED), and 655 nm (RED LED). Light pulses at different wavelengths were provided for 1 s duration every 5 s. Data were plotted using the Neural Metric tool.

**RNA-seq.** RNA was extracted using Trizol and sequenced on an Illumina HiSeq 2500 or 4000. RNA sequencing was aligned using STAR aligner[35] and quantified using Cufflinks[36].

**DNA extraction.** DNA was extracted from iPSC lines using the DNeasy blood and tissue kit (Qiagen #65904). DNA from small samples was extracted using phenol–chloroform extraction (Invitrogen #15593-031). Cells were lysed in 480 μL Lysis Buffer (10 mM Tris pH 8.0, 10 mM NaCl, and 10 mM EDTA), with 25 μL 10% SDS, and 10 μL Proteinase K for 2 h at 55 °C. In all, 5 μL RNAse was added and incubated at 37 °C for 10 min. In total, 10 μL 5 M NaCl, 1 μL glycogen, and 500 μL phenol:chloroform:isoamyl alcohol was added, and the solution was mixed by shaking. The solution was centrifuged at 16,000 rcf for 5 min. The top aqueous phase was kept and another 500 μL phenol:chloroform:isoamyl alcohol was added and mixed repeat phenol extraction. In all, 0.9 mL cold absolute ethanol was added to the aqueous phase and mixed by inverting 50 times. The solution was incubated at −20 °C for at least an hour. The solution was centrifuged 20,000 rcf for 20 min at 4 °C. The ethanol was removed, and the pellet was washed with 1 mL 70% EtOH and centrifuged for 5 min at 20,000 rcf. The wash was repeated and centrifuged at 16,000 rcf for 5 min. The pellet was air-dried and resuspended in water.

**CRISPR/Cas9 targeting of RB1.** gRNAs were targeted to the 4th Exon of RB1 (Table S5).

iPSC lines were transfected in two 20–30% confluent six-well wells using the Lipofectamine 3000 Reagent kit (ThermoFisher L3000001). Cells were transfected with both gRNAs, Cas9, and a GFP reporter. The cells were incubated for 48–72 h until 80–90% confluent. Cells were dissociated at 80% confluency by adding 1 mL of Accutase to 1 6 ww of culture and incubating at 37 °C for 3′. In total, 2 mL of mTesr was added, and the cells were combined and spun down at 300 rcf, 4′. Cells were resuspended in mTesr and sorted by flow cytometry for GFP-positive cells. The lines were purposefully left the lines mosaic (~10% insertions/deletions) so that the line would not be a complete knockout of RB1 for retinal differentiation. To quantify the indels created by the CRISPR system, DNA was extracted from the targeted iPSC lines and PCR was run using tagged primers for exon 4. Indels were quantified using CRISPResso[37].

**RNA-seq analysis.** The paired-end sequencing reads were subjected to mouse read cleansing with "bbsplit" (https://sourceforge.net/projects/bbmap/) if the sample was derived from xenografts. The adapters in sequencing reads were trimmed with "trim_galore" (v0.4.4, https://www.bioinformatics.babraham.ac.uk/projects/trim_galore/, -q 20 –phred 33 -paired). The trimmed sequencing reads were mapped with STAR[35] to the human genome GRCh38.

The expected gene counts calculated using RSEM[38] for each sample were compiled to one gene count matrix. Only genes annotated as level 1 or 2 by GENCODE (v31) were kept in the downstream analysis. In addition, only genes with count per million (CPM) more than 0.5 in at least one sample were kept. The normalization factor for each sample was calculated using "calcNormFactors" in the "edgeR" package (v3.26.8)[39], and gene expression values were transformed and normalized using voom[40] in the "limma" package (v3.40.6)[41] in R. The normalized expression values were then used to perform principal component analysis (PCA) using the "prcomp" function in R. The loadings of PC1 and PC2 were used to generate the PCA plot.

The variance of the normalized gene expression value of each gene across all samples was calculated, and the top 500 most variable genes were used for the unsupervised clustering analysis. The clustering was performed using the "complete linkage" method ("hclust" function in R) based on Euclidean distances. The heatmap ordered by hierarchical clustering results was generated using the "ComplexHeatmap" package (v2.0.0)[42] in R.

**Methylation array analysis.** A genome-wide DNA methylation array was performed with Infinium MethylationEPIC BeadChip (Illumina, CA, USA) targeting 850,000 CpG sites in accordance with the manufacturer's instructions. Raw data files generated by the iScan array scanner were read and preprocessed by using

"minfi" Bioconductor package. Unsupervised hierarchical clustering (Euclidean Distance and "Ward.D2" linkage) of DNA methylation profiling was performed based on the 5000 most variable methylation probes across all samples (or a subset of samples), which were selected by the variance of the beta values. For dimensionality reduction and visualization, PCA was performed in the initial steps using the top 5000 most variable probes and the first 50 dimensions were retained to run tSNE with perplexity values in the range (5–20) and 5000 iterations ("Rtsne" package, version 0.15, https://github.com/jkrijthe/Rtsne). Copy number variation (CNV) analysis from methylation array data was performed using the "Conumee" package (version 1.16.0). Most differentially methylated regions were detected with DMRcate (version 1.18.0)[43].

**Whole-genome sequence analysis.** The paired-end sequencing reads were mapped with bwa[44]. The in-house somatic mutation detection procedure was described previously[6]. In addition, we also used an ensemble approach to call somatic mutations (SNV/indels) with multiple published tools, including Mutect2 (v4.1.2.0)[45], SomaticSniper (v1.0.5.0)[46], VarScan2 (v2.4.3)[47], MuSE (v1.0rc)[48], and Strelka2 (v2.9.10)[49]. The consensus calls by at least two callers were considered as confident mutations. The consensus call sets were further manually reviewed for the read depth, mapping quality, and strand bias to remove additional artifacts.

In terms of somatic copy number alternations (SCNA), in addition to CONSERTING, which is described previously[50], they were also determined by CNVkit[51] and cn.Mops[52].

For somatic structural variants, five SV callers were implemented in the workflow for SV calling, including Delly (v0.8.2)[53], Lumpy (v0.2.13)[54], Manta (v1.5.0)[55], Gridss (2.5.0)[56], and novoBreak (v1.1)[57]. The SV calls passing the default quality filters of each caller were merged using SURVIVOR[58] and genotyped by SVtyper[59]. The intersected call sets were manually reviewed for the supporting soft-clipped and discordant read counts at both ends of a putative SV site using IGV.

**Single-cell RNA-sequencing analysis.** Tumor and xenograft samples were dissociated using an enzymatic tumor dissociation. The tumor was added to 10 mL of RPMI with 600 μL of trypsin (10 mg/mL, Sigma Cat#T9935) and incubated at 37 °C for 10 min. In total, 600 μL of each Soybean Trypsin Inhibitor (10 mg/mL) DNase I (2 mg/mL) and magnesium chloride (1 M) were added and the tumor suspension was filtered through a 40-μm cell strainer and centrifuged at 500 rcf. The pellet was resuspended in 5 mL of red blood cell lysis solution (5 Prime Cat#2301310) and incubated at RT for 10 min. 5 mL PBS −/− with 10% FBS was added and the cell suspension centrifuged at 500 rcf for 5 min. The supernatant was discarded, and the cell pellet was resuspended in RPMI. The cell suspension was then layered on top of a BSA cushion (4% BSA in REM) and centrifuged at 500×g for 10 min. The supernatant was removed and the cells were resuspended at ~1000 cells/μL and counted.

The human retina was obtained from the MidSouth EyeBank. The tissue was dissected and added to 400 μL papain buffer (1 mM L-cysteine with 0.5 mM EDTA in PBS−/−) with 40 U of papain. The tissue was incubated for 10–20 min at 37 °C with agitation every 5 min until dissociated. In total, 40 μL of DNase was added to the sample and it was incubated for 5 additional minutes. The dissociated cells from both methods were then passed through a 40 μM filter and washed with 3 mL REM (DMEM:F12, 10% FBS, 1% HEPES; ThermoFisher #35050-061, 1× penicillin–streptomycin; ThermoFisher #15140-122, 1% GlutaMAX, 0.05% insulin; Sigma G4386). The cell suspension was then layered on top of a BSA cushion (4% BSA in REM) and centrifuged at 500×g for 10 min. The supernatant was removed, and the cells were resuspended at ~1000 cells/μL and counted.

Approximately 10,000 cells from each sample were taken and loaded onto the 10× chromium controller for single-cell RNA-sequencing analysis which was completed according to the 10× genomics protocol. Barcoded RNA was sequenced according to 10× Genomics protocol on an Illumina HiSeq 2500 or 4000. Cell-type recognition was determined using SingleR (v1.0.1)[60], and copy number variation (CNVs) were identified using inferCNV (v1.2.1, inferCNV of the Trinity CTAT Project. https://github.com/broadinstitute/inferCNV).

In human fetal retina datasets, only cells with more than 500 genes and less than 3000 genes expressed and with <5% of mitochondrial reads are retained for analysis. The retained data were normalized and scaled using the SCTransform method[61] in Seurat 3. Dimensionality reduction and clustering were also performed using Seurat functions. A list of genes indicating S and G2M phases of the cell cycle, compiled from the cell cycle genes provided in Seurat and in the G2M human genes provided in Aldiri et al.[18], was used to identify proliferating progenitor cells. Only clusters with the enriched expression of at least three of those genes (adjusted P value <0.05, average logFC >0.5, and percent of cells in the cluster expressing the genes are 1.5-folds higher than that of the rest of the cells) were assigned as progenitor cells.

Single-cell RNA-seq reads from tumors and normal retina samples were aligned using the cell ranger pipeline (v3.0.2) to the hg19 reference data (v3.0.0). Aligned data were processed using Seurat 3. Specifically, only cells with more than 400 genes and <7000 genes expressed and with <10% of mitochondrial reads are kept for downstream analyses. All cells in normal retina samples were then merged into one dataset. Cells in normal retina and all tumor datasets were scored for their cell cycle phase using the CellCycleScoring function in Seurat 3, based on the combined

list of cell cycle genes provided in Seurat and in the G2M human genes provided in Aldiri et al.[18]. Normalization, scaling, dimensionality reduction, and clustering of the normal retina and individual tumor datasets were performed as described above, but with cell cycle effects regressed out. Cell types in the normal retina dataset were identified based on enriched expressions of signature genes.

The identified progenitor cells across all ages in the fetal retina dataset were then combined with the normal adult retina dataset to serve as a reference for the identification of cell types in tumor samples. Label transfer was performed using the default pipeline in Seurat 3 with FindTransferAnchors and TransferData functions[33].

**RNA velocity analysis**. Counts of unspliced and spliced reads from each tumor sample were derived using the velocyto command line tool[62], and reads mapped to multiple-locus or mapped inside repeat regions (derived from UCSC genome browser) were discarded. The generated loom files were then analyzed using scVelo with a generalized dynamical model[63]. Briefly, only genes with at least ten counts for spliced and unspliced RNAs were kept for velocity analysis. Normalization, modeling of transcriptional dynamics, and estimation of RNA velocities were performed using default parameters in scVelo. For visualization, single-cell velocities were projected onto the pre-computed umap embedding from Seurat.

**Reporting summary**. Further information on research design is available in the Nature Research Reporting Summary linked to this article.

## Data availability

The retinal progenitor publicly available data used in this study are available in the GEO database under accession code GSE116106. The sequencing data generated in this study have been deposited in the GEO database under accession codes GSE174200, GSE174201, and GSE174202. The processed single-cell data are available in a Cloud-based viewer (https://pecan.stjude.cloud/static/rbsinglecell). The remaining data are available within the Article, Supplemental Information, or Source Data file.

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

## Acknowledgements

Grant funding to M.A.D. from the National Institutes of Health (CA245508, EY030180), the Shipley Foundation, Alex's Lemonade Stand Foundation supported this research and to J.L.N. (CA225442). The project was also supported by NIH grant CA21765 and American Lebanese Syrian Associated Charities. We thank Mark and Virginia Valentine in the Cytogenetics Shared Resource for karyotyping and FISH and Shondra Miller in the Center for Advanced Genome Engineering for designing the gRNAs. We thank Vickie Given and Julie Overbey for clinical research support and Matt Lear for the coordination of sample collection. We thank Anand Patel for assistance with single-cell RNA-seq analysis and Jake Dundee for technical assistance. The MidSouth eye bank provides normal adult retina and Anita Bhattacharyya at WiCell for generation of iPSC lines. The content is solely the responsibility of the authors and does not necessarily represent the official views of the National Institutes of Health.

## Author contributions

J.L.N., A.N., R.B., and M.A.D. conceived and designed the study, R.B. acquired patient samples, J.L.N., A.N., L.G., Q.T., S.A., J.Z., and K.N. collected the data, J.L.N., K.L., X.C., H.J., and G.W. performed computational analysis, J.L.N., A.N., K.L., X.C., M.W., E.S., H.J., G.W., B.O., Q.T., D.J., R.B., and M.A.D. analyzed and interpreted the data. C.R. and X.Z. developed visualization tools and a web portal. M.A.D. drafted the manuscript.

## Competing interests

The authors declare no competing interests.
