## [Peer Review File · Nature Communications]

REVIEWERS' COMMENTS

Reviewer #1 (Remarks to the Author):

The authors have adequately addressed all of our previous comments/concerns with the exception of our comments regarding Figure 2E. It is still not clear what the point of the MEA responses of a single d175 3D-RET-derived organoid is. There appears to be robust response to 475 nm and 530 nm but little to no response to 655 nm. What is the interpretation for this? Line 156 states that Fig 2E is further demonstration that the 3D-RET method is indistinguishable from that of Sasai, but no comparable recordings from d175 Sasai method-generated organoids are shown. Also, although it may be difficult and time-consuming to repeat the experiments on new organoids, there needs to be some demonstration that the 3D-RET technique generates convincing outer segments (by ICC or light microscopy if not EM) in order to include this data.

Reviewer #2 (Remarks to the Author):

I reviewed a previous version of this manuscript. All of my concerns have been adequately addressed by the authors. I look forward to seeing this work in print.

Reviewer #3 (Remarks to the Author):

The authors carefully addressed the points raised in the first round of reviews at Nature. It is now easier to appreciate the improvements in the 3D retinal organoids protocols. The ability to grow retinoblastoma from normal cells in germline patients will provide new ways to investigate mechanisms of tumor initiation and progression in a human context. The single-cell RNA-seq data provide a nice resource for the field. Finally, the identification of hybrid gene expression in

retinoblastoma cells may help understand tumor plasticity upon loss of RB. Overall, this is a great study, with very useful resources for the field.

REVIEWERS' COMMENTS

Reviewer #1 (Remarks to the Author):

The authors have adequately addressed all of our previous comments/concerns with the exception of our comments regarding Figure 2E. It is still not clear what the point of the MEA responses of a single d175 3D-RET-derived organoid is. There appears to be robust response to 475 nm and 530 nm but little to no response to 655 nm. What is the interpretation for this? Line 156 states that Fig 2E is further demonstration that the 3D-RET method is indistinguishable from that of Sasai, but no comparable recordings from d175 Sasai method-generated organoids are shown. Also, although it may be difficult and time-consuming to repeat the experiments on new organoids, there needs to be some demonstration that the 3D-RET technique generates convincing outer segments (by ICC or light microscopy if not EM) in order to include this data.

We would like to thank reviewer #1 for their comments and for acknowledging the time it would take to address their concerns. We have therefore removed the multi-electrode array from the manuscript. Still, to address the question of why there is little response at 655 nm, we believe it may be due to 655 nm being at the edge of the human visible spectrum and therefore the cells of the organoids may be too immature to respond or not express the correct opsin to respond at that wavelength.

Reviewer #2 (Remarks to the Author):

I reviewed a previous version of this manuscript. All of my concerns have been adequately addressed by the authors. I look forward to seeing this work in print.

We thank reviewer #2 for their comments and their time and effort reviewing our manuscript.

Reviewer #3 (Remarks to the Author):

The authors carefully addressed the points raised in the first round of reviews at Nature. It is now easier to appreciate the improvements in the 3D retinal organoids protocols. The ability to grow retinoblastoma from normal cells in germline patients will provide new ways to investigate mechanisms of tumor initiation and progression in a human context. The single-cell RNA-seq data provide a nice resource for the field. Finally, the identification of hybrid gene expression in retinoblastoma cells may help understand tumor plasticity upon loss of RB. Overall, this is a great study, with very useful resources for the field.

We thank reviewer #3 for their comments and their time and effort reviewing our manuscript.